# The Effect of Pythagorean Self-Awareness on Heart Rate Variability, Perceived Stress and Behavior of Preschool Children

**DOI:** 10.3390/children9101529

**Published:** 2022-10-06

**Authors:** Kyriaki Angelopoulou, Eleni Zaverdinou, Flora Bacopoulou, George P. Chrousos, Giorgos Giannakakis, Christina Kanaka-Gantenbein, Sophie Mavrogeni, Maria Charalampopoulou, Maria Katimertzi, Christina Darviri

**Affiliations:** 1Postgraduate Course of Stress Management and Health Promotion, School of Medicine, National and Kapodistrian University of Athens, 15771 Athens, Greece; 2Center for Adolescent Medicine, First Department of Pediatrics, School of Medicine, National and Kapodistrian University of Athens, 15771 Athens, Greece; 3First Department of Pediatrics, Medical School, National and Kapodistrian University of Athens, Aghia Sophia Children’s Hospital, National and Kapodistrian University of Athens, 15771 Athens, Greece; 4UNESCO Chair in Adolescent Health Care First Department of Pediatrics, School of Medicine, National and Kapodistrian University of Athens, 15771 Athens, Greece; 5Unit of Clinical and Translational Research in Endocrinology, First Department of Pediatrics, School of Medicine, National and Kapodistrian University of Athens, 15771 Athens, Greece; 6Foundation for Research and Technology—Hellas, 70013 Crete, Greece; 7Onassis Cardiac Surgery Center, 17674 Athens, Greece

**Keywords:** stress, kindergarten, Pythagorean Self-awareness, stress management, heart rate variability, photoplethysmography, preschool

## Abstract

Stress is associated with unhealthy habits and non-communicable diseases. It is also linked to communicable diseases due to its impact on immune function. These can be prevented through intervention programs in schools. The aim of this study was to examine the effect of the simplified Pythagorean Self-Awareness Intervention on heart rate variability (HRV) parameters, perceived stress and behaviors of preschool children. The sample of the study consisted of 45 preschool students. A “one group (double) pretest—posttest design” was used, to allow for comparisons of the measurements before and after the intervention. Students were assessed via two questionnaires (“Perceived Stress Scale for Children” (PSS-C) and “Checklist for Screening Behavioral Problems in Preschool Children”) and a photoplethysmographic (PPG) device. The intervention lasted 9 weeks and included practicing of the Pythagorean Self-awareness techniques and the adoption of healthy behaviors. The results showed no statistically significant differences between the two pretests (*p* > 0.05 for all comparisons) and statistically significant differences between the second pretest and posttest (“Perceived Stress Scale for Children”, (PSS-C) *p* < 0.0001, “Checklist for Screening Behavioral Problems in Preschool Children” *p* < 0.0001 and two indices of PPG device: heart rate mean, *p* < 0.0001, low frequency/very low frequency, *p* = 0.034). In conclusion, the Pythagorean Self-Awareness Intervention had a beneficial effect on the sample of preschool students examined, as the results showed an improvement in the perceived stress and the HRV parameters tested, and in engaging healthier behaviors, findings that indicate a relaxed psychologic state and a healthier lifestyle.

## 1. Introduction

Studies have shown that experiencing stressful situations in fetal age, childhood and adolescence is particularly damaging and appears to be linked to the later onset of diseases. When someone faces a stressful stimulus, the autonomic nervous system (ANS) is activated. More specifically, for an individual to cope, the sympathetic system is activated, which, among other things, is responsible for increasing heart rate and breathing. Therefore, the function of the cardiovascular and respiratory systems can act as a biomarker of stress. Measurements of blood pressure in children have shown its correlation with stress. Studies have shown the importance of measuring heart rate (HR) and heart rate variability (HRV) and their correlation to stress [1].

More specifically, in stress response, cardiovascular tone and respirations are increased, pro-inflammatory cytokines are released, endothelial function is temporarily reduced and platelets are activated. Consequently, secondary effects related to the cardiovascular and respiratory systems may represent important indicators of chronic stress in children [2]. Accordingly, high variability in heart rate indicates a healthy and adaptive autonomic nervous system, while low variability may be an early indicator of abnormal ANS function [3]. The activation of the ANS occurs after the exposure of the individual to a stressful situation. More specifically, the autonomic nervous system and the hypothalamus-pituitary-adrenal axis are activated leading to the secretion of hormones, mainly cortisol [4].

Stress can affect both adults and children. It is possible for the constant daily friction of stressful events to result in the development of chronic stress. Its early recognition is considered essential to deal with, with the least possible psychosomatic strain of the person experiencing this condition. It is common for people, whether they are adults or children, to perceive as stressful events not the objective risks, but the subjective ones. Therefore, the physiological symptoms of stress can be caused after the perceived stress [5]. This in children causes emotional and behavioral problems. According to parental reports, children were strongly led to introversion after being exposed to events they perceived as intense and stressful. This is fully explained through the child’s perceived stress in everyday life [6]. According to literature, continuous stressful situations or experiencing a significant stressful event, can change the neurobiological systems of children, undermining their health, their adaptation to society as well as the control of their emotions and their behavior [7].

Nowadays, the fast pace of everyday life and the activities which children choose, may be harmful to them (e.g., using screens), intensify stressful, and lead them to face serious problems. The way they spend their free time affects their body, brain and behavior. A difference in caloric intake and energy expenditure is observed, violent behavior is adopted and beneficial activities are neglected. Screen use may cause obesity, metabolic syndrome, aggressive behavior and mental disorders [8]. Finally, parents’ lifestyle, coping with difficulties, methods of educating their children and their habits have an impact on the development, behavior, school performance and their lifestyle. It is observed that people who have experienced significant changes in their family environment, such as death or absence of parents or any other change, often experience intense stress, and adopt harmful behaviors which affect their mental and physical health even in the long term. Studies have shown that estranged families suffer anxiety and depression which affects the child’s emotional regulation, development and integration into society [9].

One way to avoid the negative effect of stress is Pythagorean Self-Awareness Intervention (PSAI). It is a stress management technique based on the principles of the Greek philosopher and mathematician Pythagoras. It involves the practice of abdominal breathing and the 12 Pythagorean virtues: cooperation, order and accuracy, fairness, truthfulness, industriousness, discipline and respect for the law, contribution to the general good, courage, decent behavior, cleanliness, temperance, charity and spirituality. It also involves the adoption of a healthy lifestyle. Everyone who practices this technique needs to recall the recent events every day and night and evaluate them according to healthy eating, daily exercise, sleep, screen time, social relationships and the 12 virtues through three questions: “What have I done right?”, “What have I done wrong?” and “What have I omitted that I ought to have done?” In this way, self-control, self-evaluation and a healthy lifestyle are taught. According to the literature, the PSAI has a positive physiological and psychological impact on children and adults [4,10,11,12,13,14,15,16,17,18,19,20] which will be further analyzed in the next sections. The aim of this study was to examine the effect of Pythagorean Self-awareness on heart rate variability, perceived stress and behaviors, in preschool children.

## 2. Materials and Methods

### 2.1. Sample

This study was conducted at the “Rodokipos” Nursery and Kindergarten in Marousi, Attica, Greece, during the school hours, with the presence of school staff. Consent was given by the school administration. Firstly, information about the program that would be implemented at the school was given to the parents, followed by their written consent for the participation of their children.

The eligible sample of the study consisted of all the healthy students of the “Rodokipos” Nursery School and Kindergarten, which equals to 50 children. The inclusion criteria were defined as the age group of 3–6 years as well as the ability to speak and draw. Exclusion criteria were the diagnosis of a mental health problem, difficulty in the ability to speak or muteness and the practice of other stress management techniques. As a result, five children were excluded. Two of those were unable to participate in the program and three were absent from school for a long period of time during which the program was implemented. The remaining 45 children comprised the study sample.

### 2.2. Study Design

Even though the population of kindergartens is small, it is not possible to use participants of several different school units, to avoid confounding factors. Additionally, since the specific field of research is addressed to children, it was considered necessary for the entire sample to participate in the program for ethical reasons. For the same reasons, the sample was not divided into a control and intervention group but was used as one group whose measurement results were compared before and after the intervention, i.e., one group pretest—posttest design. With the aim to strengthen the specific study’s design, two measurements were carried out initially (double pretest). According to this strategy, if the difference between either pretest or posttest is much greater than the difference between the two pretests, additional support is provided for the effect of the specific factor. This validates that any change after the intervention, comes from the specific program instead of any other factor [21].

### 2.3. Measurement Procedure

In this research, the first pretest was carried out 2 months before the intervention, the second 1 month before the intervention and the posttest 1 month after it. Each time, the participants were given two questionnaires. The first test, named “Perceived Stress Scale for Children” (PSS-C), was filled out by the children and the second, named “Checklist for Screening Behavioral Problems in Preschool Children”, was filled out by the kindergarten teachers. Afterwards, a photoplethysmography (PPG) device was used. Each child took a comfortable position, in a quiet room of the kindergarten, in the presence of the two researchers, while two plethysmographs were placed on its two index fingers, recording its heart rate for 5 min. The measurement procedure for all children was completed in 5 days.

#### 2.3.1. Perceived Stress Scale for Children (PSS-C)

This scale is a very popular tool regarding the measurement of psychological stress and chronic anxiety. It consists of 14 questions. Each one can be answered on a scale consisting of 4 options (never—a little—sometimes—a lot). A higher value on the total score suggests higher stress perception (the first question is not included in the score and questions 3, 6, 7, 10, 11, 13 and 14 are reversed scored). This scale assesses the extent to which individuals feel that their lives have been unpredictable, uncontrollable, and overloaded in the past month. The items assessed are general in nature and do not focus on specific events or experiences. This scale was first used in 2014 in 153 children (5–18 years old) with anxiety disorders in the United States [22]. Since then, it has been used globally. For example, it has been used in a study on perceived stress among Japanese students during the COVID-19 quarantine [23], in a study of perceived stress in relation to school performance before and after physical activity in South African primary school children [24], in a study of the lifestyles of children in the developing world [25], in a study of arts therapy for reducing stress in children in the United States [26] and in a research on stress reduction and management in US elementary school students [27].

#### 2.3.2. Checklist for Screening Behavioral Problems in Preschool Children

This tool is used for the valid and reliable identification of behavior problems in preschool children. It is filled out by teachers and consists of 25 questions. Twenty-four of them refer to forms of behavior and the last question refers to the kindergarten teacher’s assessment of each child’s behavior. These questions address 4 key aspects of the child’s recent behavior related to concentration, conduct, emotional difficulties and social relationships. A child showing deviant behavior to a severe degree will receive a score of 2, while a moderate degree of deviant behavior will lead to a score of 1. Finally, a child exhibiting normal behavior for its age will receive a score of 0. In questions 1 and 18, the child receives a score of 2 in the first and last answers, as those two are judged to be abnormal. In question 6, the second and fourth answers receive the grade 0, as they are considered normal. Question 24 is scored according to the degree of frequency of occurrence of even one of the listed behaviors. The child with the lowest total score is closer to normal behavior. This questionnaire was first used in 2005 in 220 children (4–6 years old) in kindergartens located in Crete island, Greece [28]. Since then, it has been used in many studies, for the assessment of children’s behavior problems in relation to their gender and origin in the school environment, for the social and emotional assessment of infants and toddlers [29], for the correlation between maternal eating disorders and nutritional problems of children [30] and for the behavior of preschool children in England [31].

#### 2.3.3. PPG Stress Flow Device

This medical device (Bio Tekna Co., Via Pialoi, Venice, Italy) is used to monitor and record autonomic nervous system activity and heart rate variability.

It has received approval from the Italian Ministry of Health. It is also CE certified as a non-invasive medical device for diagnostic and monitoring purposes. The device consists of two plethysmographs; one for the index finger of the right hand and one for the left index finger. Their sensors consist of a light—emitting diode (LED) and an optical receiver, which receives the transmitted light of the LED. The examinee takes a comfortable position with the hands placed on the surface of a table. The sensors are placed on the fingers to record the small changes in light caused by the heartbeats (according to the principles of photoplethysmography). In this way, the HR mean, total power, low frequency/very low frequency (LF/VLF), very low frequency (VLF), low frequency (LF) and high frequency (HF) indexes, were measured. The duration of this process is 5 min [32].

### 2.4. Intervention

The intervention was carried out for 4 h in the morning, once a week, during school hours. Children were divided into 2 groups according to their age (22, 3–4 years old children and 23, 5–6 years old children) and each time they were trained at first in groups and then individually. During the rest of the week, a further analysis of what was learned during the intervention, was carried out. The total duration of the intervention was 9 weeks. Before the start of the first meeting of the intervention, the children were individually trained in abdominal breathing, using a biofeedback device. In this intervention, students were taught a specific routine that they were asked to apply at school and at home. Each meeting included children’s abdominal breathing exercises, an interview of the students and a discussion of their experiences related to the program. They were asked about lifestyle habits (exercise, diet, sleep, screen time and behavior at home and at school). These data were noted by the researchers (qualitative data). Then, a different topic for each week was presented. To consolidate each topic, a fairy tale was read, and a video was presented. During each week, the students were engaged in activities and making crafts related to each topic. The children were provided with the “Student’s Notebook” created to review all the topics. They were also provided with the “Self-awareness Diary”, in which the events of each student’s day were noted, as well as their daily habits.

In the first two weeks, the main point of the intervention was the teaching of the Pythagorean self-awareness technique. This is a stress management technique that teaches self-control, self-evaluation, adoption of a healthy lifestyle and enhances memory. According to this, every morning and every night before they went to sleep, children had to be positioned at their beds in a relaxed manner. Then, they had to practice abdominal breathing for 10 min. After that, they had to review all the events of the day from the moment they woke up until the moment they went to bed. The children should evaluate the events of the day and their habits related to exercise, nutrition, sleep, screen time and social relationships. They had to ask themselves 3 questions: “What have I done right?”, “What have I done wrong?” and “What have I omitted that I ought to have done?”. Children should also repeat and judge themselves according to the practice of the 12 Pythagorean virtues: cooperation, order and accuracy, fairness, truthfulness, industriousness, discipline and respect for the law, contribution to the general good, courage, decent behavior, cleanliness, temperance, charity and spirituality. The purpose of the evaluation through the 3 questions and the 12 virtues was the teaching of self-awareness. The children should reward their right actions and correct the wrong ones. As a result, they gradually gained self-control and healthy lifestyle.

In the third week, the children learned about healthy eating. They adopted a fixed meal schedule. They started to choose foods from all food groups and especially fruits and vegetables. They adjusted to drinking 8 glasses of water per day and avoiding sweets, candies and fast food.

In the fourth week, the children learned about social relationships and bullying. The students learned the importance of respecting and embracing diversity. They started to be polite to all their classmates, teachers and family. They also learned to manage and express their feelings. They tried to be helpful to their school and to their home.

In the fifth week, students learned about the importance of sleep. They adopted a sleep schedule to rest and be healthy. They learned that they should always sleep at 8 o’ clock and wake early in the morning.

In the sixth week, students learned that screen time should not exceed a duration of 1 h per day. They also learned about the effect of screens on sleep. Screens should be turned off at least 2 h before bedtime.

In the seventh week, students learned about the impact of physical activity and exercise on health. They were encouraged to wear pedometers and take 12,000 steps per day. Children began to engage in sports, dance and other activities.

In the eighth and ninth week, children created their daily schedule, which included the healthy habits they had been taught. They reviewed all the topics and discussed the experiences they gained from the program. At the end of the intervention, a “Health festival” was held, where students shared with parents, teachers and researchers their achievements due to the program.

### 2.5. Statistical Analysis

Questionnaire answers and PPG Stress Flow data were collected. The statistical analysis was performed using SPSS v26.0 (IBM, New York, NY, USA). Data are presented as frequencies (%) for categorical variables and as mean (Standard Deviation, SD) and median (interquartile range, IQR) for categorical variables. Normality of quantitative variables was checked using the Shapiro–Wilk test. For the statistical evaluation of the quantitative variables of the pretests and the quantitative variables before and after the intervention, the non-parametric Wilcoxon Signed Ranks Test was performed. The significance level was set at 0.05 for all analyses.

## 3. Results

### 3.1. Demographic Characteristics of the Sample

Table 1 presents the demographic characteristics of the sample. In total, 25 (55.6%) children were boys and 20 (44.4%) were girls. The mean age (SD) was 4.12 (0.98) years.

### 3.2. Comparison between the Two Pretests

Table 2 shows the results of the two initial pretests, which include two questionnaires (Perceived Stress Scale for Children and Checklist for Screening Behavioral Problems in Preschool Children) and the indices of PPG device. Specifically, the listed values are mean, median, SD, IQR of quantitative variables, as well as *p*-value. Comparison of participants’ baseline measurements (double pretests) shows that there was no statistically significant difference (*p* > 0.05 for all comparisons).

### 3.3. Comparison of the Pretest and Posttest Results

The participants’ behavior before and after the intervention (pretest—posttest) was also investigated using the second measurement (i.e., before the intervention) and for the last measurement (i.e., after the intervention).

The questionnaires’ quantitative variables Perceived Stress Scale (Perceived stress PSS-C, *p* < 0.0001, effect size r = 0.65 and confidence interval, lower bound: −4.59, upper bound: −1.91) and the Checklist for Screening Behavioral Problems in Preschool Children (Behavior Checklist, *p* < 0.0001, effect size r = 0.79 and confidence interval, lower bound: −5.31, upper bound: −2.96) are presented in Table 3. It can be observed that after the intervention, there was a statistically significant reduction in the questionnaires (*p* < 0.05) and a large effect size (r > 0.5).

Moreover, the HRV parameters in time and frequency domain were extracted using the PPG device. The HRV statistical results are presented in Table 4.

It can be observed that after the intervention, there was a statistically significant reduction in the heart rate (*p* < 0.001), LF/VLF (*p* = 0.034). The pretest—posttest effect size with its confidence interval can also be observed. Regarding the HR mean, effect size r was large (0.70) with its confidence interval (lower bound: −9.42, upper bound: −4.70). Regarding the LF/VLF, effect size r was small (0.32) with its confidence interval (lower bound: −0.05, upper bound: −0.001). The total power, VLF, LF and HF did not present any statistical difference, which strengthens r with its confidence interval.

## 4. Discussion

The aim of this study was to examine the impact of PSAI on heart rate variability parameters, perceived stress and behaviors in preschool children. The results showed a reduction in perceived stress, as indicated by the results of the “Perceived Stress Scale for Children”. Moreover, improvement in anger, guilt, emotion management, shyness and social relationships was found by the “Checklist for Screening Behavioral Problems in Preschool Children”. Finally, a reduction in HR mean was also found.

The HR index indicates the activity of the autonomic nervous system. Specifically, the activation of the sympathetic system, which occurs during a stressful situation, is accompanied by an increase in heart rate. On the contrary, the activation of the parasympathetic system, which occurs after a stressful stimulus, indicates a relaxed state, accompanied by a decrease in heart rate. HRV, the physiological phenomenon of variation in the time interval between heartbeats, is primarily dependent on the HR index. Normally, HR and HRV are increased in children compared to adults. As age increases, HR and HRV decrease. According to the literature, increased HR and decreased HRV values can be correlated to stress [33,34]. These differences indicate activation of parasympathetic system, which leads to stress reduction and a relaxed state. HRV and HR are associated with lifestyle behaviors and disease diagnosis, such as aerobic exercise, sleep, drug use, smoking and alcohol drinking, as well as diagnosis of obstructive sleep apnea, myocardial infarction, myocardial ischemia and disorders of the nervous system [35,36,37]. The results of the present study showed an impact on lifestyle behavior and a reduction in perceived stress which, agree with the above-presented literature.

HRV parameters include HF which has been used as an index of parasympathetic activity. Another index, LF has been associated with both sympathetic and parasympathetic systems, but it is still debated in the field of psychophysiology. Some researchers suggest that LF is a measure of sympathetic modulation [38] while others argue that it is a measure of both sympathetic and parasympathetic modulation [39] and others consider it to represent baroreflex activity [40]. The above suggests that in some contexts, LF may serve as an index of sympathetic modulation, while in other contexts, there may be extensive parasympathetic influence. Previous studies relating to the autonomic nervous system or HPA axis to perceived stress have yielded inconsistent results that may be partially attributable to measurement limitations due to the young age of the study population. According to the literature, less is known about the impact of VLF. Studies have shown correlation with thermoregulation and the renin–angiotensin system, as well as metabolic and hormonal influences [41]. This study showed that two HRV indices were improved, while others may need further research, as the literature so far is poor.

Improvement of children’s behaviors was also observed by quality data which were noted after the intervention. Phrases such as “I feel calm”, “I sleep better, without having nightmares”, “I prefer to eat vegetables rather than candies”, “I exercise everyday”, “I drink 8 glasses of water every day” and “I reduced screen time” were quoted.

The beneficial effect of the Pythagorean Self-Awareness Intervention has been shown in other studies. According to them, it effects declarative memory, working memory and the Default Mode Network. In this way, it improves memory and school performance. It also reduces stress and anxiety, emotional eating and improves weight management, acne vulgaris, sleep and lifestyle behaviors. It is has also been studied in patients with non-communicable diseases, such as breast cancer, diabetes mellitus type 2, multiple sclerosis, depressive disorder and chronic insomnia [4,10,11,12,13,14,15,16,17,18,19,20]. All of the above factors, justify the effect of the Pythagorean Self-Awareness Intervention in preschool children, which is documented for the first time.

According to the literature, there is a lack of studies on holistic stress-management programs implemented on preschool children. Usually, the conducted programs focus on healthy diet, exercise, weight control, hygiene, emotional intelligence and stress control separately [42,43,44]. In older children, programs have been conducted about exercise, nutrition, weight management, addiction, healthy lifestyle, school bullying, safe internet use and stress management, using the techniques of abdominal breathing, progressive muscle relaxation, mindfulness and Cognitive Behavioral Therapy (CBT) [45,46,47,48,49,50,51,52,53].

The Pythagorean Self-Awareness Intervention taught the children about stress reduction, abdominal breathing and engagement in a healthy lifestyle. They learned about right and wrong to correct their mistakes and become self-controlled individuals. They asked themselves three questions: “What have I done right?”, “What have I done wrong?” and “What have I omitted that I ought to have done?”. Children repeated and judged themselves according to the 12 Pythagorean virtues. They adopted a lifestyle which included healthy diet, daily exercise, sleep routine, screen time reduction and healthier social relationships. These achievements justify a reduction in HR and perceived stress. Children were in a relaxed state and were engaged with healthier behaviors.

The field of research combining holistic programs and preschool children is limited. Although in this study there was an effort to standardize the real-life experimental conditions, there remains a questionable validity regarding the very young age of the participants. In addition, there are few studies addressing the investigation of self-awareness through HRV parameters and we hope that an increased sample as well as other reported studies will enhance the repeatability of this research question under investigation.

Future research in a larger sample is thus considered important, as well as the use of questionnaires about stress management and lifestyle, created especially for preschool children.

## 5. Conclusions

Stress is a very important issue as it is linked to an unhealthy lifestyle and diseases. It affects not only adults, but also children shaping their lives. It is, therefore, considered necessary to implement stress management and health promotion programs in schools for prevention.

In the present research, a holistic program for preschool children was carried out for the first time. The aim of this study was to exam the effect of the Pythagorean Self-Awareness Intervention on HRV parameters, perceived stress and behavior on preschool children. Children were trained on abdominal breathing, daily exercise, healthy eating, scheduled sleep and screen time as well as better social relationships.

According to the literature, HRV indicates the activation of the autonomic nervous system and can be used as a biomarker of stress. It is associated with lifestyle behaviors and diagnosis of diseases. Its reduction indicates improvement in lifestyle, stress management and a relaxed state. Moreover, the Pythagorean Self-Awareness Intervention impacts memory and the Default Mode Network. It is responsible for stress reduction, improvement in diseases and adoption of healthier habits.

During the Pythagorean Self-awareness Intervention, children adopted self-control and a healthier lifestyle. The results showed an improvement in HRV indices, perceived stress and the adoption of healthier behaviors. The results were verified by biomarkers, two questionnaires and quality data. More specifically, two HRV parameters (heart rate mean, *p* < 0.0001, low frequency/very low frequency, *p* = 0.034) were improved. Furthermore, perceived stress as validated by the questionnaires (“Perceived Stress Scale for Children” (PSS-C) *p* < 0.0001 and “Checklist for Screening Behavioral Problems in Preschool Children” *p* < 0.0001) was decreased. Finally, healthier behaviors were adopted as shown by the acquired qualitive data. Further research using larger sample and questionnaires created especially for preschool children is needed to verify our findings.

## Figures and Tables

**Table 1 children-09-01529-t001:** Demographic characteristics of the sample.

Sex Ν (%)	Sample N = 45
Boys	25 (55.6)
Girls	20 (44.4)
**Age**	
Mean (SD)	4.12 (0.98)
Median (IQR)	4.00 (2.00)

**Table 2 children-09-01529-t002:** Initial pretest results.

Questionnaires and PPG Indices	1st Pretest Mean (SD) Median (IQR)	2nd Pretest Mean (SD) Median (IQR)	*p*-Value
PSS-C	6.69 (5.34)	5.44 (4.61)	0.210
	5.00 (7.00)	4.00 (7.00)	
CSBPPC	4.87 (4.13)	4.87 (4.13)	1.000
	5.00 (7.00)	5.00 (7.00)	
HR mean	97.04 (10.45)	98.64 (9.34)	0.103
	95.70 (14.45)	97.60 (12.60)	
Total Power	8.10 (1.03)	8.14 (0.92)	0.950
	8.08 (1.27)	8.14 (1.17)	
LF/VLF	1.03 (0.09)	1.08 (0.08)	0.08
	1.03 (0.16)	1.08 (0.12)	
VLF	6.61 (0.85)	6.56 (0.83)	0.955
	6.52 (1.09)	6.52 (1.18)	
LF	6.85 (1.04)	6.99 (0.91)	0.269
	6.72 (1.32)	6.96 (1.25)	
HF	7.23 (1.30)	7.27 (1.08)	0.739
	7.24 (1.74)	7.26 (1.49)	

PSS-C = Perceived Stress Scale for Children; CSBPPC = Checklist for Screening Behavioral Problems in Preschool Children; HR = heart rate; LF/VLF = low frequency/very low frequency; VLF = very low frequency; LF = low frequency; HF = high frequency.

**Table 3 children-09-01529-t003:** Comparison of the pretest and posttest results.

Measurements before and after Intervention Mean (SD) Median (IQR)					
	Pretest Mean (SD) Median (IQR)	Posttest Mean (SD) Median (IQR)	*p*-Value	Effect Size r	95% Confidence Interval Lower Bound Upper Bound
PSS-C	5.44 (4.61)	2.24 (2.96)	<0.0001	0.65	−4.59
	4.00 (7.00)	0.00 (4.00)			−1.91
CSBPPC	4.87 (4.13)	0.78 (1.95)	<0.0001	0.79	−5.31
	5.00 (7.00)	0.00 (0.00)			−2.96

PSS-C = Perceived Stress Scale for Children; CSBPPC = Checklist for Screening Behavioral Problems in Preschool Children.

**Table 4 children-09-01529-t004:** The HRV statistical results for the pretest and posttest condition.

HRV Feature	Pretest Mean (SD) Median (IQR	Posttest Mean (SD) Median (IQR)	*p*-Value	Effect Size r	95% Confidence Interval Lower Bound Upper Bound
HR_mean_	98.64 (9.34) 97.60 (12.60)	91.69 (9.09) 92.10 (11.47)	<0.0001	0.70	−9.42 −4.70
Total power	8.14 (0.92) 8.14 (1.17)	7.93 (0.99) 7.91 (1.4)	0.171	0.21	−0.52 0.11
LF/VLF	1.08 (0.08) 1.08 (0.12)	1.05 (0.08) 1.04 (0.12)	0.034	0.32	−0.05 −0.001
VLF	6.56 (0.83) 6.52 (1.18)	6.50 (0.82) 6.60 (0.98)	0.744	0.05	−0.35 0.22
LF	6.99 (0.91) 6.96 (1.25)	6.78 (0.93) 6.69 (1.34)	0.193	0.20	−0.53 0.11
HF	7.27 (1.08) 7.26 (1.49)	6.91 (1.32) 6.72 (1.71)	0.073	0.27	−0.73 0.01

HR = heart rate; LF/VLF = low frequency/very low frequency; VLF = very low frequency; LF = low frequency; HF = high frequency.

## Data Availability

Data available on request due to restrictions, e.g., privacy or ethical.

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
