# Peer review of "The Effect of Pythagorean Self-Awareness on Heart Rate Variability, Perceived Stress and Behavior of Preschool Children"

_children, 2022, doi:10.3390/children9101529_

Round 1
Reviewer 1 Report
In my own opinion, this paper had contributed to the literature and highlighted further the importance of childcare. In particular, the study had validated further the new instruments used for detecting and managing stress among children. The research design is also easy to follow and replicated, as well as the descriptive statistical analysis and presentation of results.
However, because the instruments helped to assess the impact of the Pythagorean Self-awareness Intervention (PSI) I think a paragraph or more may be devoted to explaining the PSI in the introduction section. This would serve as a cohesive link to the next section, instead of waiting to read about the PSI in the intervention subsection.
Can the authors reframe the conclusion section? I believe they can do better. Fine examples of how to make conclusion sections to reflect research papers, and how research papers achieved their goals, are available in international journals.
Author Response
Dear Reviewer 1,
We would like to thank you for giving us the opportunity to improve our manuscript. Your comments have helped us clarify aspects of our work and communicate better the reported results.
Please find below a detailed point-by-point response letter to your comments.
Thank you again for your comments.
Yours sincerely,
Kyriaki Angelopoulou
“However, because the instruments helped to assess the impact of the Pythagorean Self-awareness Intervention (PSI) I think a paragraph or more may be devoted to explaining the PSI in the introduction section. This would serve as a cohesive link to the next section, instead of waiting to read about the PSI in the intervention subsection.”
A paragraph explaining the PSAI has been added in the introduction section.
“Can the authors reframe the conclusion section? I believe they can do better. Fine examples of how to make conclusion sections to reflect research papers, and how research papers achieved their goals, are available in international journals.”
The conclusion section has been reframed.
Reviewer 2 Report
This is an interesting study, and my comments are largely limited to the minor editorial issues listed below. Because effect size is commonly reported in the clinical literature, report of the pretest-posttest effect size with its confidence interval would help connect these results with other investigations of heart rate variability.
I concur with the authors in expressing the hope for a larger multisite follow on study that would incorporate a control group. One speculates that the Perceived Stress Scale for Children might be particularly vulnerable to learning effects. This possibility emphasizes the importance of psychophysiological biomarkers.
Minor Editorial Points
Abstract “stress is associated with unhealthy habits and non-communicable diseases.” Could it also be argued that stress compromises immune function and increases vulnerability to communicable disease?
In several places reference is made to “the autonomous nervous system.” “Autonomic nervous system” would be the more conventional reference.
Page 4. “4 hours in the morning.” Was this every day or one day a week?
Were all student coached in self-awareness in a single group of 45, or was instruction presented to multiple smaller groups?
Page 2. “and design through stationery.” With apologies, I don’t understand what is meant.
Author Response
Dear Reviewer 2,
We would like to thank you for giving us the opportunity to improve our manuscript. Your comments have helped us clarify aspects of our work and communicate better the reported results.
Please find below a detailed point-by-point response letter to your comments.
Thank you again for your comments.
Yours sincerely,
Kyriaki Angelopoulou
“Because effect size is commonly reported in the clinical literature, report of the pretest-posttest effect size with its confidence interval would help connect these results with other investigations of heart rate variability.”
Effect size and confidence interval have been added.
“Abstract “stress is associated with unhealthy habits and non-communicable diseases.” Could it also be argued that stress compromises immune function and increases vulnerability to communicable disease?”
The association of stress to communicable diseases has been added in abstract.
“In several places reference is made to “the autonomous nervous system.” “Autonomic nervous system” would be the more conventional reference.”
The phrase “Autonomous nervous system” has been corrected to “Autonomic nervous system”.
“Page 4. “4 hours in the morning.” Was this every day or one day a week?”
The exact hours of the intervention have been added.
Were all student coached in self-awareness in a single group of 45, or was instruction presented to multiple smaller groups?
The groups of the children have been analyzed (page 4).
Page 2. “and design through stationery.” With apologies, I don’t understand what is meant.
The phrase “and design through stationary” has been corrected (page 2).